# Unraveling the role of salt-sensitivity genes in obesity with integrated network biology and co-expression analysis

Jamal Sabir M. Sabir[1,2], Abdelfatteh El Omri[1,2], Babajan Banaganapalli[3,4], Nada Aljuaid[2], Abdulkader M. Shaikh Omar[5], Abdulmalik Altaf[6], Nahid H. Hajrah[1,2], Houda Zrelli[1,2], Leila Arfaoui[7], Ramu Elango[3,4], Mona G. Alharbi[5], Alawiah M. Alhebshi[5], Robert K. Jansen[1,8], Noor A. Shaik[3,4], Muhummadh Khan[1,2]*

1 Center of Excellence in Bionanoscience Research, King Abdulaziz University, Jeddah, Saudi Arabia, 2 Genomics and Biotechnology Section and Research Group, Department of Biological Sciences, Faculty of Science, King Abdulaziz University, Jeddah, Saudi Arabia, 3 Department of Genetic Medicine, Faculty of Medicine, King Abdulaziz University, Jeddah, Saudi Arabia, 4 Princess Al-Jawhara Center of Excellence in Research of Hereditary Disorders, King Abdulaziz University, Jeddah, Saudi Arabia, 5 Biology, Department of Biological Sciences, Faculty of Science, King Abdulaziz University, Jeddah, Saudi Arabia, 6 Department of Surgery, Faculty of Medicine, King Abdulaziz University, Jeddah, Saudi Arabia, 7 Clinical Nutrition Department, Faculty of Applied Medical Sciences, King Abdulaziz University, Jeddah, Saudi Arabia, 8 Department of Integrative Biology, University of Texas at Austin, Austin, TX, United States of America

* mkmkhan@kau.edu.sa

**Data Availability Statement:** All relevant data are within the paper and its Supporting Information files.

## Abstract

Obesity is a multifactorial disease caused by complex interactions between genes and dietary factors. Salt-rich diet is related to the development and progression of several chronic diseases including obesity. However, the molecular basis of how salt sensitivity genes (SSG) contribute to adiposity in obesity patients remains unexplored. In this study, we used the microarray expression data of visceral adipose tissue samples and constructed a complex protein-interaction network of salt sensitivity genes and their co-expressed genes to trace the molecular pathways connected to obesity. The Salt Sensitivity Protein Interaction Network (SS$^{PIN}$) of 2691 differentially expressed genes and their 15474 interactions has shown that adipose tissues are enriched with the expression of 23 SSGs, 16 hubs and 84 bottlenecks (p = 2.52 x 10–16) involved in diverse molecular pathways connected to adiposity. Fifteen of these 23 SSGs along with 8 other SSGs showed a co-expression with enriched obesity-related genes (r ≥ 0.8). These SSGs and their co-expression partners are involved in diverse metabolic pathways including adipogenesis, adipocytokine signaling pathway, renin-angiotensin system, etc. This study concludes that SSGs could act as molecular signatures for tracing the basis of adipogenesis among obese patients. Integrated network centered methods may accelerate the identification of new molecular targets from the complex obesity genomics data.

## Introduction

Obesity, an excessive body fat accumulation in individuals acts as a major risk factor for the development of diverse chronic diseases like impaired insulin metabolism, glycemic

**Funding:** This project was funded by the Deanship of Scientific Research (DSR), King Abdulaziz University, Jeddah, KSA, under grant no. HiCi-63-130-35. The authors are thankful to DSR for the technical and the financial support.

**Competing interests:** The authors have declared that no competing interests exist.

**Abbreviations:** BC, Betweenness Centrality; BIND, Biomolecular Interaction Network Database; BioGRID, Biological General Repository for Interaction Datasets; DC, Degree Centrality; DEGs, differentially expressed genes; DIP, Database of Interacting Proteins; FDR, False Discovery Rate; GWAS, Genome-wide association studies; HPRD, Human Protein Reference Database; MINT, The Molecular Interaction database; PCC, Pearson's correlation algorithm; PPIM, Protein Interaction Map; RMA, Robust Multiarray Average; SSG, Salt sensitivity genes; SS$^{PIN}$, Salt Sensitivity Protein Interaction Network.

abnormalities, hypertension and cardiovascular diseases in future. Obesity, owing to its complex multifactorial disease nature is not only challenging the molecular scientists to decode its molecular basis but also the clinicians who are involved in treating, preventing and disease management. Approximately 30% of the world population is either overweight or obese [1]. So far, the specific molecular and cellular mechanisms through which environmental factors increase the risk of developing obesity in genetically susceptible individuals still remains to be a mystery. The chronic low inflammation in different tissues is one of the characteristic features of obesity [2]. Particularly, chronic inflammatory reactions which takes place in adipose tissues contribute to the obesity associated insulin insensitivity. Adipose tissue plays an important role in the development of metabolic diseases due to dysregulated discharge of adipocytokines from adipocytes in visceral fat of obese individuals. This will subsequently induce insulin resistance condition in muscles and liver. The faulty insulin sensitivity of adipose tissues, connects the obesity with other chronic diseases like diabetes, hyperlipidemia, arthritis, hypertension, cardiovascular disease, ischemic stroke, hyperglycemia and different types of cancer [3] [4].

The importance of excess salt intake in the pathogenesis of metabolic diseases is widely recognized. Salt sensitivity is a physiological trait, in which the changes in salt intake parallel the changes in blood pressure [5]. The gene expression status of salt sensitivity genes (SSGs) in adipose tissues is not yet well explored. In the present study, we focused on SSGs expressed in adipose tissues to figure their influential role in the pathogenesis of obesity. We considered genes from renin-angiotensin system pathway which maintains the homeostasis of salt and body fluids, and regulate the blood pressure [6]. In addition, expression of renin-angiotensin system in adipose tissue is involved in the regulation of triglyceride accumulation, adipocyte formation, glucose metabolism, lipolysis, and the initiation of the adverse metabolic consequences of obesity [7], [8]. Therefore, in order to identify the candidate genes from SSGs and their molecular signature networks connected to the pathogenesis of obesity, the gene expression datasets collected from visceral adipose tissues were analyzed by knowledge based systemic investigations and statistical methods. We used different statistical parameters like graph theory to pick up biomarkers from the gene expression data. We also used gene-gene correlation, which relies on the fact that disease candidate genes showing a similar expression pattern are more likely to interact with one another for their biological functioning[9]. Our network biology integrated investigation will offer novel association with potential biological comprehensions and supports future translational assessment on SSGs and obesity.

## Materials and methods

### Gene expression dataset

The microarray generated gene expression dataset with the reference ID of GSE88837 was collected from GEO (Gene Expression Omnibus) database [10]. This gene expression data is generated on Affymetrix microarray platform using the total RNA extracted from human visceral adipose tissue of 16 overweight woman adolescent samples (BMI > 25) and 14 lean adolescent women (BMI < 25). Complete information about the individuals and testing methods, can be found in **S1 Table**.

### Normalization of gene expression data

Gene expression data analysis of the samples were implemented by means of R packages [11] [12]. For the standardization and noise reduction in the probe data, CEL files were incorporated into R package, Affy, and the unprocessed signal intensity values of each gene expression probe sets were standardized with help of a statistical algorithm called as RMA (Robust

Multiarray Average). This RMA algorithm performs the normalization of raw intensity data by generating a matrix of gene expression data whose background is corrected and log2 conversion, and then quantile normalization was performed [12]. The standardized samples were then quantitatively categorized as normal (control) and obese (disease) sets. The statistical difference between differentially expressed genes (DEGs) was computed using unpaired t-test measure among healthy and obese samples [13]. For examine the statistical differences in DEGs, the false discovery rate of Benjamini and Hochberg with a p value of 0.05 was conducted[14].

## Building network of proteinprotein interaction

Bisogenet, a cytoscape plugin, was used to derive associations between the DEGs obtained from the profiles of expression. Bisogenet finds significant gene interactions from high-performance experiments and deposited literature data in DIP (Database of Interacting Proteins), BIND (Biomolecular Interaction Network Database), BioGRID (Biological General Repository for Interaction Datasets), MINT (The Molecular Interaction database), HPRD (Human Protein Reference Database), and IntAct databases [13, 15–19].

## Construction of subnetwork

The complex interactome Protein Interaction Network (PIN) was rescaled to a significant subnetwork of Salt Sensitivity Protein Interaction Network (SS$^{PIN}$) by following admitted notions in the network biology. From the Protein Interaction Network, we extracted genes that belong to (a) hubs based on degree centrality (DC), (b) betweenness centrality (BC) based bottlenecks (c) salt sensitivity genes. The PIN created from Bisogenet was optimized and imported to Cytoscape 3.2.1 in order to represent and measure the different parameters like DC and BC connected to network centrality of each individual protein in the biological network [20]. The Network Analyzer [21] Cytoscape plugin was deployed to monitor the network's local and global centrality parameters [14, 22–24].

## Selection of hub proteins

DC of a gene is the number of partners that are connected to that specific gene. Genes which shows higher DC in any given biological network will possess many interacting partners[25]. In PIN, genes having higher DC corresponds to essential genes. For identifying the hubs, we followed the hub classification approach, which was previously described by Rakshit et al., [26]. The cut-off scores used for DC, while selecting the hub protein is described as:

$$Hubs = Avg(DC) + [2 \times SD(DC)] \qquad \text{(Formula 1)}$$

where *Avg* is the average DC of significantly expressed genes in the PIN and SD denotes the standard deviation values [26].

## Identification of bottlenecks

The higher DC is in correspondence to biologically essential genes, but DC is unable to quantify significance of any gene in a network [27]. Based on the theory of the protein's local property, DC does not assess the global value of the protein in the network. There could be several other key indicators that show the importance of a protein in the network based on its global significance. A global BC measure was therefore implemented to determine the characteristics of any query gene at the entire interactome level [28]. BC is measured by applying following

formula:

$$BC\left(n\right) = \sum\nolimits_{s \neq n \neq t} \left[\frac{\sigma_{st}(n)}{\sigma_{st}}\right] \qquad \text{(Formula 2)}$$

where 's' and 't' are the network nodes, other than 'n' and $\sigma_{st}(n)$ is the number of shortest paths from $s$ to $t$ that 'n' lies upon [29]. The significantly expressed genes falling in top 25% are regarded under bottleneck category using the node betweenness distribution.

## Salt sensitivity genes

The genes involved in the pathway of renin angiotensin aldosterone system were collected as they serve as chief component in the regulation of salt and water balance of the body [30]. We also collected salt sensitivity genes from a detailed literature survey [31] [5] [32] [33]. In total, we obtained 47 SSG as represented in **S2 Table.**

## Mapping of weighted gene-gene correlations

The map detailing gene-gene correlations was created on the basis of the algorithm known as the Pearson correlation across the entire gene set in the SS$^{PIN}$. The "r" value indicating the correlation between gene pairs in the expression data was generated with help of Pearson's correlation coefficient (PCC) method. The formula used for calculating PCC for gene pairs is described in below given Formula 3.

$$PCC\left(r\right) = \frac{\sum_{i=1}^{n}\left(x_i - \underline{x}\right)\left(y_i - \underline{y}\right)}{\sqrt{\sum_{i=1}^{n}\left(x_i - \underline{x}\right)^2}\sqrt{\sum_{i=1}^{n}\left(y_i - \underline{y}\right)^2}} \qquad \text{(Formula 3)}$$

where $\underline{x}$ and $\underline{y}$ indicates average of the expression values of two genes in the samples, respectively.

## Functional enrichment analysis

Functional enrichment analysis validates the physiological importance of the genes involved in a biological process and helps to reveal unintended gene activity. ToppGene Suite was employed to perform functional enrichment of the filtered genes [34].

# Results

## Microarray gene expression profile analysis

We obtained 2691 significant genes from the analysis of raw gene expression signals using RMA with statistical significance of p value $\leq$ 0.05. The intensity values of genes in the expression profiles, before and after normalization, are depicted as box plots which represents standardized form of representing the data distribution in Fig 1.

## Constructed Protein Interaction Network

Overall 2691 differentially expressed genes generated from the microarray expression profile were inputted in Bisogenet, a plugin in Cytoscape, to create PIN by extracting all potential connectivity between the genes. The created PIN comprised of outliers like replicated edges and self-loops. The PIN is transformed to a stable network by eliminating self-loops and replicated edges which is then used to calculate the standardized graph centrality parameters for each single gene. The plugin created a complex PIN, covered of 2691 nodes and 15474 edges with edge-node ratio of

**Before normalization**

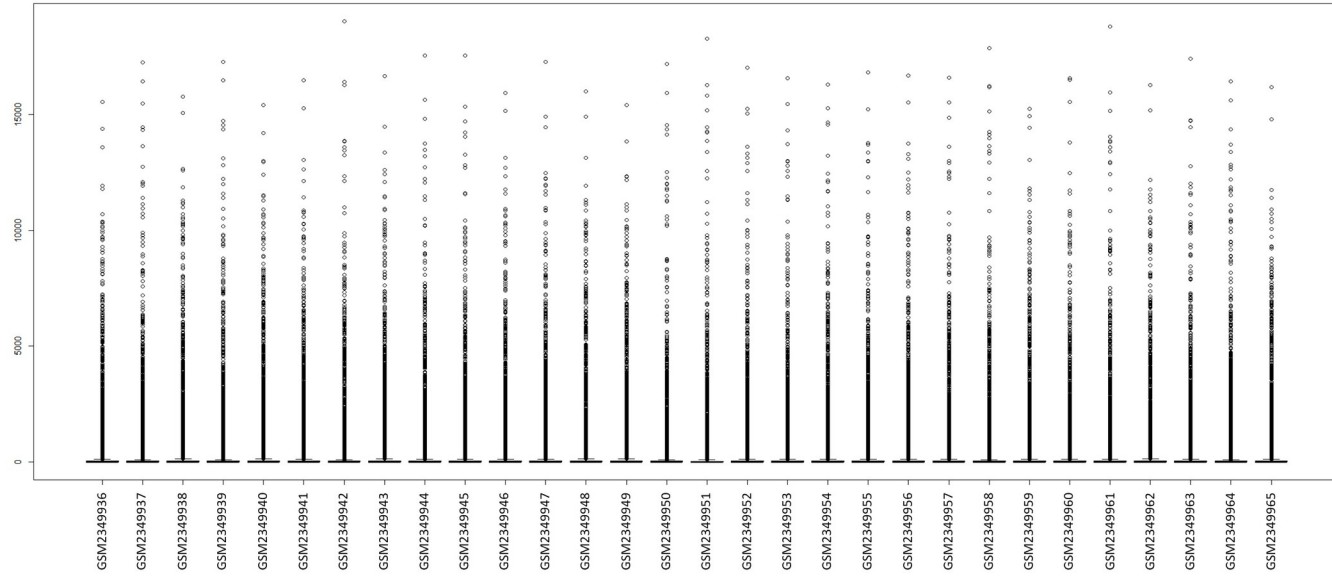

**After normalization**

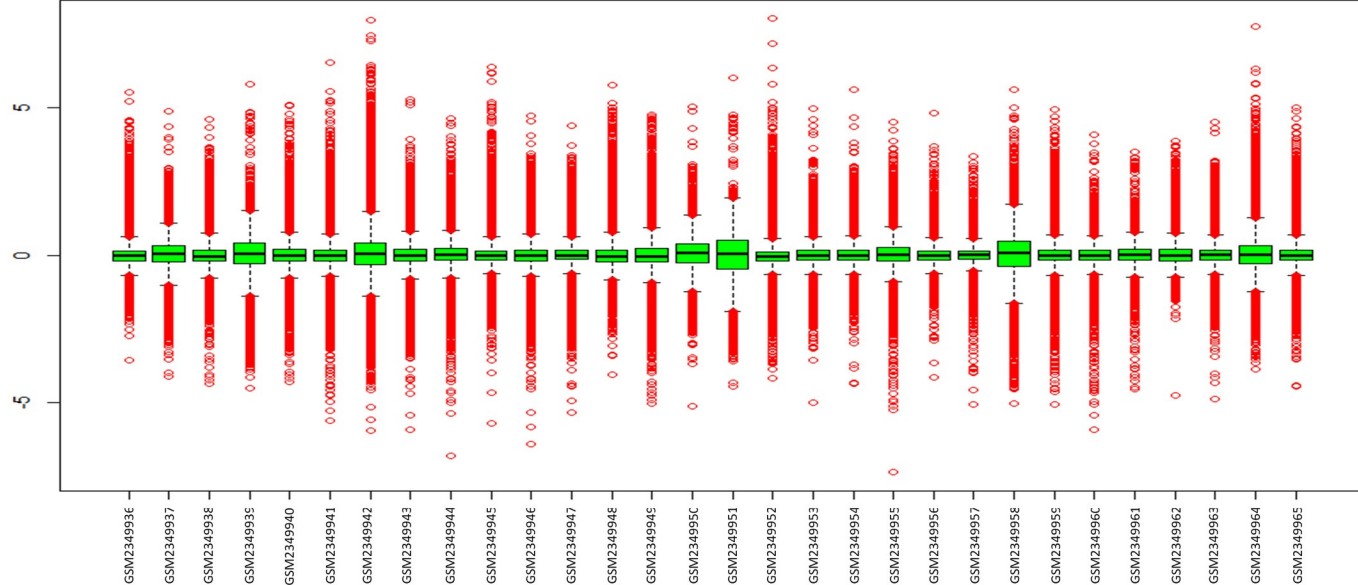

**Fig 1. Pre and post-normalization of microarray gene expression data.** Samples are represented on horizontal axis and the gene expression values on vertical axis.

5.75 on an average. Next, the plugin NetworkAnalyzer, calculated the degree centrality betweenness centrality parameters of the network which are considered as local and global graph parameters respectively [21]. **Table 1** provides a description of the top 10 significant genes dependent on the highest degree centrality along with general parameters of centrality.

## Salt Sensitivity Protein Interaction Network (SS$^{PIN}$)

PIN genes have been grouped into hubs and bottlenecks based on criteria of graph centrality to establish a large network of protein interactions. The cut-off limit for hubs and bottlenecks

**Table 1. List of 10 significant genes obtained from network analysis based on graph theory.**

| Gene | Name | BC# | DC$ |
|------|------|-----|-----|
| PHF8 | PHD Finger Protein 8 | 0.260 | 882 |
| EGR1 | Early Growth Response 1 | 0.091 | 504 |
| JUND | Jund Proto-Oncogene, Ap-1 Transcription Factor Subunit | 0.089 | 462 |
| FOS | Fos Proto-Oncogene, Ap-1 Transcription Factor Subunit | 0.076 | 438 |
| CHD2 | Chromodomain Helicase Dna Binding Protein 2 | 0.051 | 371 |
| APP | Amyloid Beta Precursor Protein | 0.070 | 366 |
| IRF1 | Interferon Regulatory Factor 1 | 0.036 | 306 |
| STAT3 | Signal Transducer And Activator Of Transcription 3 | 0.049 | 295 |
| TEAD4 | TEA Domain Transcription Factor 4 | 0.042 | 292 |
| RELA | RELA Proto-Oncogene, Nf-Kb Subunit | 0.039 | 278 |

#BC = Betweenness Centrality

$DC = Degree Centrality

was defined in the methods section on the basis of Formulas 1 & 2. The degree of the hubs ranged from 86 to 882 nodes which makes an average connectivity of 208 edges per node. We obtained 40 hubs, 502 bottlenecks and 47 SSGs. 15 of 47 SSGs were also found to act as bottleneck in the interactome (**Table 2**). Hubs, bottlenecks and SSGs were together consisted of 574 genes with 5356 interactions. For the ease of exploration genes in hubs, bottlenecks and SSGs were grouped as HBS. The interaction among these 574 genes in HBS were mapped from PIN to create new network of Salt Sensitivity Protein Interaction Network.

## Functional enrichment analysis

We used the ToppGene computational annotation system to determine the functional and biological importance of the genes. The genes of HBS have been enriched by 2192 biological

**Table 2. The salt sensitivity genes overlapping with bottleneck genes.**

| Symbol | Name | BC# | DC$ |
|--------|------|-----|-----|
| ADD1 | Adducin 1 | 0.001 | 17 |
| ADRB2 | Adrenoceptor beta 2 | 0.00305 | 62 |
| AGT | Angiotensinogen | 0.00105 | 16 |
| AGTR1 | Angiotensin II receptor type 1 | 0.00097 | 9 |
| ATP6AP2 | ATPase H+ transporting accessory protein 2 | 0.00102 | 20 |
| CYP11B2 | Cytochrome P450 family 11 subfamily B member 2 | 0.00084 | 4 |
| GNAI2 | G protein subunit alpha i2 | 0.00283 | 31 |
| GNB3 | G protein subunit beta 3 | 0.00041 | 10 |
| MME | Membrane metalloendopeptidase | 0.00047 | 22 |
| NEDD4L | Neural precursor cell expressed, developmentally down-regulated 4-like, E3 ubiquitin protein ligase | 0.00216 | 46 |
| PRCP | Prolylcarboxypeptidase | 0.00052 | 9 |
| PREP | Prolyl endopeptidase | 0.00048 | 11 |
| SCNN1A | Sodium channel epithelial 1 alpha subunit | 0.00043 | 12 |
| SGK1 | Serum/glucocorticoid regulated kinase 1 | 0.00149 | 32 |
| WNK1 | WNK lysine deficient protein kinase 1 | 0.00096 | 33 |

#BC = Betweenness Centrality

$DC = Degree Centrality

**Table 3. The genes involved in obesity categorized as hubs, bottlenecks and salt sensitivity genes.**

| Category | Genes | P-value |
|---|---|---|
| Salt sensitivity genes | ACE, ACE2, ADD1, ADRB2, AGT, AGTR1, AGTR2, ANPEP, ATP6AP2, CMA1, CYP17A1, GNB3, GRK4, KLK1, LNPEP, MAS1, MME, NEDD4L, PRCP, PRKG1, REN, SGK1, TH | $2.52 \times 10^{-16}$ |
| Hubs | EGR1, JUND, FOS, APP, STAT3, JUN, STAT1, ATF3, SIRT7, FOXM1, TBL1XR1, BAG3, HSPB1, CEBPD, HNRNPA1, VCAM1 | $2.52 \times 10^{-16}$ |
| Bottlenecks | CALM1, PCNA, JUNB, CRK, SHC1, GAPDH, WWOX, ITCH, HSPA1A, CRY2, NFKB1, MLH1, PKM, HSPD1, PTPN11, MAP1LC3B, TUFM, APC, SNRNP200, CDK5, CALR, HLA-C, GTF2I, PRKAR1A, BCL2L1, TNF, IGF1R, ZFP36, NR4A1, TANK, SOD2, KRT18, JAK3, SMARCA2, NUP62, PRKCZ, DNMT1, ATG5, DNAJB1, STAT5B, LEPR, VDR, PIK3CA, PPIA, FOXO3, MYD88, CAST, DDAH2, VEGFA, SOCS3, PINK1, COL1A1, THBS1, ACAT2, THRA, SNAP29, VTI1B, PER1, TPI1, RGS2, BMPR1A, NPHP1, FTL, GTF2H1, APOE, CYCS, ABCA1, CSK, TIMM44, GNAQ, C3, POLDIP2, SLU7, ST13, COL4A1, LAMA1, SDC2, IGF1, BGN, CFH, ADM, WASF1, HGF, C1QTNF6 | $2.52 \times 10^{-16}$ |

processes (BP), 210 molecular functions (MF), 246 cellular components (CC), 642 pathways and 1669 diseases. Analysis of enrichment accounted for about 125 obesity genes. The obesity related genes consisted of 23 SSGs, 16 hubs and 84 bottleneck genes. Approximately 50% of the SSGs were observed to be involved in obesity via functional enrichment analysis (**Table 3**). These genes were also involved in pathways associated with obesity like *regulation of lipolysis in adipocytes*, *adipogenesis*, *adipocytokine signaling pathway*, *renin-angiotensin system*, *signaling by leptin*, *toll-like receptor pathway*, *PI3K-Akt signaling pathway*, *ras signaling pathway*, *cytokine signaling in immune system insulin pathway*, *glucocorticoid receptor regulatory network* and *NF-kappa B signaling pathway* (**Table 4**). The detailed list of genes involved in these pathways are given in the **S3 Table**.

The enriched genes were also involved in obesity related diseases like Diabetes Mellitus, Hypertensive disease, Asthma, Autoimmune Diseases, Diabetes Mellitus (Insulin-Dependent), Congestive heart failure, Cardiovascular Diseases, Coronary Artery Disease, Heart failure, Coronary heart disease, Coronary Arteriosclerosis, Depressive disorder, Hyperglycemia, Metabolic Syndrome X, Essential Hypertension, Ischemic stroke, Hyperlipidemia. Obesity is one of leading cause of aforesaid diseases. The interaction map of genes to the diseases is depicted in the Fig 2.

## Co-expression analysis

The expression pattern similarity between 574 HBS genes was established and ranked based on Pearson's correlation algorithm (Fig 3) for array of control and disease samples. For control and disease samples (Formula 3), the algorithm created *PCC* for 328329 pair of genes from 574 genes. Gene pairs were screened in this approach based on established concepts such as i) gene expression level with high positive correlation. ii) Genes with similar patterns of speech are more likely to interact. In obesity studies, gene pairs with value r = 0.8 are chosen from the correlation map as higher r score indicates a greater relationship. Corresponding gene pairs were extracted from normal correlation map to identify the variation in the co-expression from obesity to normal sample. Totally, 226 genes are observed to co-express with obesity related genes with 1126 interactions in obesity condition (Fig 4). There were 88 obesity related genes and 23 SSGs in the set which were co-expressed in samples of obese adipose tissue. We focused on the 23 SSGs that are found to have co-expressed with obesity related genes.

By performing co-expression analysis, we obtained 23 co-expressed SSGs with obesity related genes. Eight among the 23 co-expressed genes were not previously reported for the

**Table 4. The enriched pathways that are closely associated with obesity or obesity related diseases.**

| Pathway | P-value | Source | Gene's Count |
|---|---|---|---|
| Adipocytokine signaling pathway | 1.86E-02 | KEGG | 7 |
| Adipogenesis | 5.56E-03 | Wikipathways | 12 |
| Cellular responses to stress | 7.36E-06 | REACTOME | 39 |
| Chemokine signaling pathway | 4.22E-04 | KEGG[2] | 18 |
| Cytokine Signaling in Immune system | 9.58E-08 | REACTOME | 62 |
| Glucocorticoid receptor regulatory network | 1.45E-07 | PID | 16 |
| Hemostasis | 7.39E-04 | REACTOME | 43 |
| Insulin signaling pathway | 6.39E-05 | PID[1] | 9 |
| Interferon Signaling | 2.18E-06 | REACTOME | 24 |
| MAPK signaling pathway | 6.88E-04 | KEGG | 22 |
| Mineralocorticoid biosynthesis | 4.78E-03 | BIOCYC | 2 |
| NF-KB signaling pathway | 1.12E-03 | BioCarta | 5 |
| NOD-like receptor signaling pathway | 5.32E-04 | KEGG | 17 |
| PI3K-Akt signaling pathway | 8.92E-06 | KEGG | 32 |
| Ras signaling pathway | 3.61E-04 | KEGG | 21 |
| Regulation of lipolysis in adipocytes | 1.16E-03 | KEGG | 8 |
| Renin secretion | 2.61E-04 | KEGG | 10 |
| Renin-angiotensin system | 3.25E-30 | KEGG | 22 |
| Signaling by Leptin | 8.10E-03 | REACTOME | 19 |
| Signaling by Rho GTPases | 2.63E-03 | REACTOME | 30 |
| Sodium/Calcium exchangers | 7.92E-03 | Reactome | 3 |
| Sphingolipid signaling pathway | 2.94E-03 | KEGG | 12 |
| TNF signaling pathway | 1.46E-06 | KEGG | 17 |
| Toll-like receptor pathway | 2.23E-03 | BioCarta | 6 |
| Type I diabetes mellitus | 2.61E-02 | KEGG | 5 |
| Type II diabetes mellitus | 8.93E-03 | KEGG | 6 |

[1]PID = pathway interaction database

[2]KEGG = Kyoto Encyclopedia of Genes and Genomes

disease obesity via functional enrichment analysis. The list of co-expressed SSGs are depicted in the Table 5. We developed an interaction map of unreported SSGs with obesity related genes (Fig 5) by taking their co-relation score as weight (**Table 6**). We extracted the edge weight of gene pairs in both obese and normal sample to identify the distinct variations across set of two conditions. This attempt was performed because of the fact that differentially co-expressed genes participate in numerous biological processes resulting in adverse or complementary effects.

It is very clear from the plot that majority of the co-expressed genes in the obese conditions are not co-expressed in normal conditions. Considering them as a disease subnetwork, we calculated the local topological parameters based on graph theory. Among 8 unreported SSGs, the highly connected genes with obesity related genes is ENPEP followed by WNK1. These two genes were having 21 and 20 direct connectivity to the obesity related gene in the co-expressed state. The SSGs, THOP1, CLCNKB, SCNN1G and THOP1 were having poor connectivity in the disease subnetwork. Notably, CYP3A5 and CTSA formed two separate networks with connectivity 6 and 3 respectively to obesity related gene. The interactions of the unreported SSGs with obesity related genes was separated and depicted in the Fig 6. We have narrowed down unreported SSGs to 5 prioritized genes (ENPEP, WNK1, CYP3A5, SLC24A3 and CTSA) based on their co-expression and topological parameters.

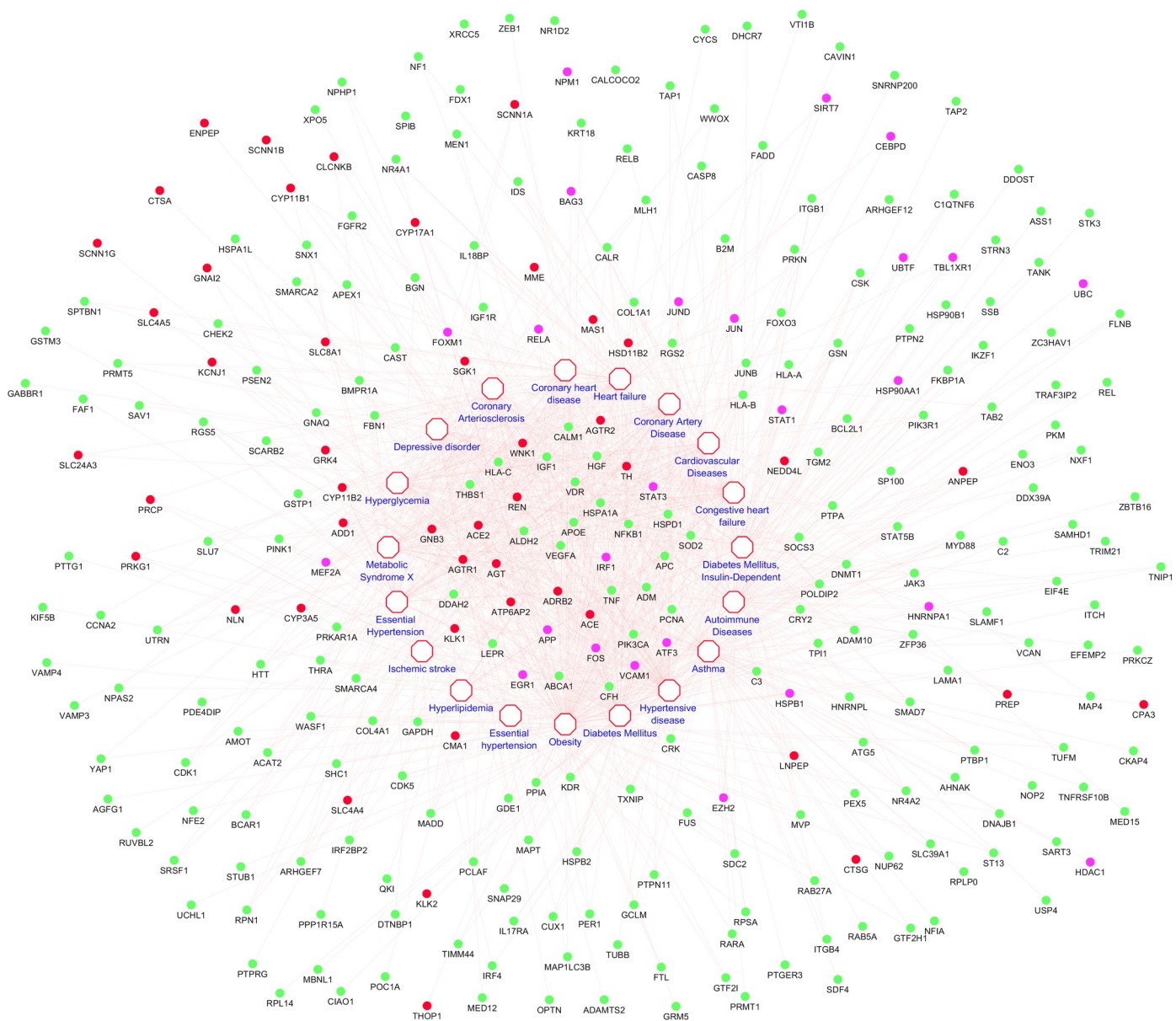

**Fig 2. The interaction map of disease to genes.** The red nodes represents salt sensitivity genes, pink and green nodes represents hubs and bottlenecks respectively.

An attempt was made to associate novel genes found in this study to the genome wide association studies on many disease traits from around the world in the GWAS catalog (MacArthur et al., 2016). We extracted the reported traits of these co-expressed genes from GWAS catalog to identify their association to obesity (Table 7). Many of these traits were related to obesity or its associated traits in cardiovascular or metabolic diseases.

## Discussion

Traditional gene profiling approaches are based on detecting individual targeted genes showing variations in the experimental group versus the control one. However, mere identification of differentially expressed genes cannot always help in understanding biological pathways

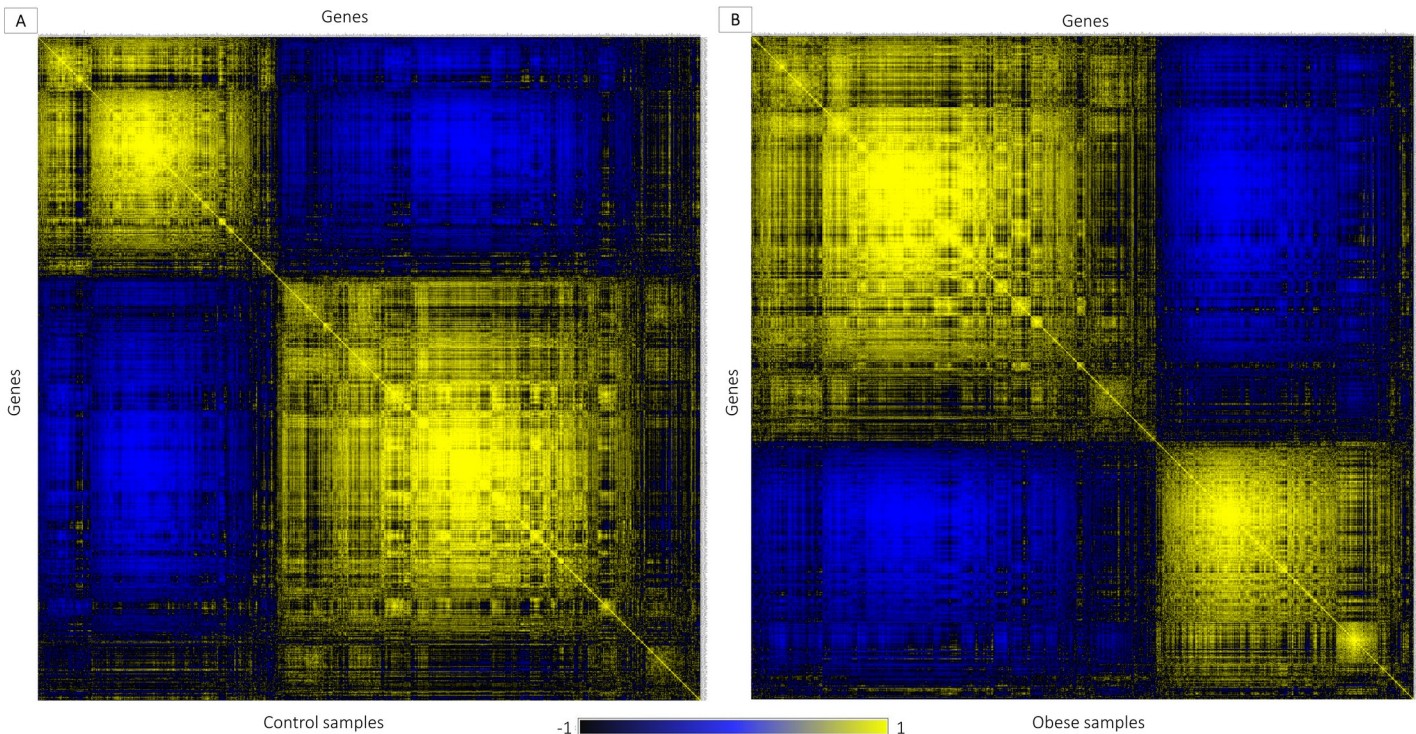

**Fig 3. Representation of gene-gene correlation plot.** The correlation plots illustrate substantial variations in gene expression among the gene pairs in the control (lean) and obese samples. A). Gene-gene correlation of lean samples (control), B). Gene-gene correlation of obese samples (disease)

(metabolism, transcription, and gene interactions, among others) regulations involved in the disease pathogenesis [35]. This is especially true in case of multifaceted or complex disorders like obesity, which do not progress because of instabilities in a single gene, but due to the changes in several pathways comprising of various biological networks [14]. In the current study, we investigated the concepts of gene regulatory networks in order to profile the significant variations of salt-sensitive genes involved in obesity.

Local parameter DC and global parameter BC were used to dissect the complex interactome. DC of a gene is the number of partners that are connected to that specific gene. Protein Interaction Network (PIN) are mathematical representations of physical and/or functional interaction between nodes, where nodes are the genes and the edges represent the connection between them, which may be binding possibility, metabolic interaction or regulatory crosstalks [36]. In our built PIN, significant alterations were observed in the expression level of h selected genes in our experimental settings. Initially, a complex network of significant genes from adipose tissue was constructed which was further decomposed to a Salt Sensitivity Protein Interaction Network based on hubs and bottlenecks. Hubs are considered as key features in networks, because they project critical intersections, which gets disturbs the networks whenever they are removed [37]. In the constructed interactome PIN, highly essential genes show high degree of connectivity. Several publications strongly suggested that diseased genes have higher connectivity and cross-talks when compared to non-diseased ones which are supporting hubs impact in the network [38]. We obtained 40 hubs with an average connectivity of 208 edges. The enrichment analysis revealed that 16 hub genes were involved in obesity and 13 hubs were involved in Type 2 Diabetes, closely related to obesity. Thus, the identification of hub molecules in the PIN is of substantial interest to get better insights of the disease pathogenesis. On other hand, functionally relevant vertices (nodes) in the network were detected

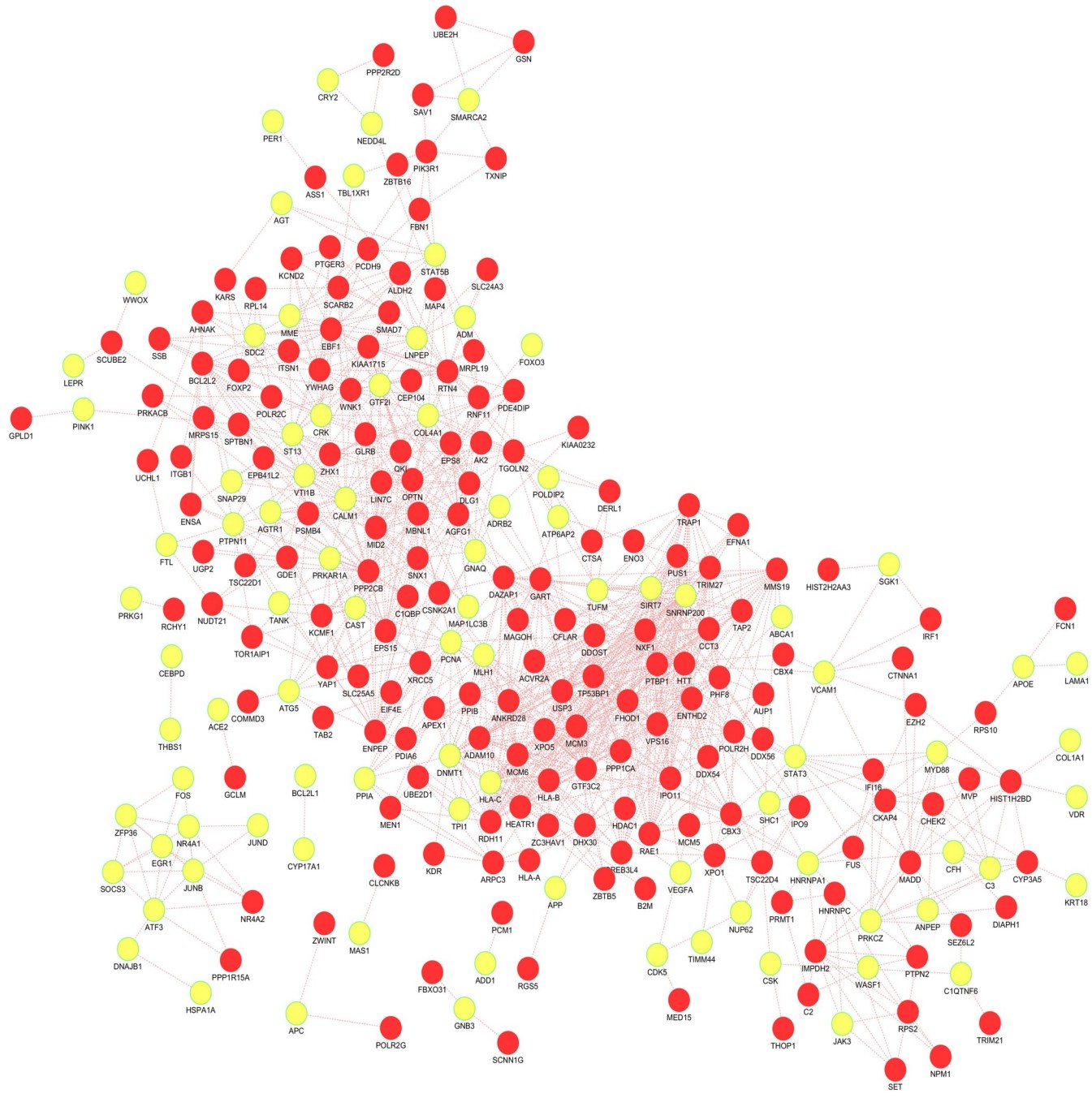

**Fig 4. The plot of genes co-expressed with obesity related genes.** The obese condition where yellow nodes represents obesity related genes.

using betweenness centrality (BC). In fact, This approach helped to sort-out vertices linking dense networks, rather than nodes located inside the dense cluster[28]. Functional enrichment analysis represented 84 bottleneck genes in obesity.

The unintended interactions of the genes may lead to deregulated functions. Hence, to better understand the gene function in cellular context, we need to understand how genes are interconnected together within several biological processes and molecular signaling pathways. In fact this type of structural and functional bio interactome can be created by evaluating the

**Table 5. List of co-expressed salt sensitive genes with their identity in obesity.**

| Gene | Name | Role in obesity |
|---|---|---|
| ACE2 | Angiotensin I Converting Enzyme 2 | Reported |
| ADD1 | Adducin 1 | Reported |
| ADRB2 | Adrenoceptor Beta 2 | Reported |
| AGT | Angiotensinogen | Reported |
| AGTR1 | Angiotensin Ii Receptor Type 1 | Reported |
| ANPEP | Alanyl Aminopeptidase, Membrane | Reported |
| ATP6AP2 | Atpase H+ Transporting Accessory Protein 2 | Reported |
| CYP17A1 | Cytochrome P450 Family 17 Subfamily A Member 1 | Reported |
| GNB3 | G Protein Subunit Beta 3 | Reported |
| LNPEP | Leucyl And Cystinyl Aminopeptidase | Reported |
| MAS1 | Mas1 Proto-Oncogene, G Protein-Coupled Receptor | Reported |
| MME | Membrane Metalloendopeptidase | Reported |
| NEDD4L | Neural Precursor Cell Expressed, Developmentally Down-Regulated 4-Like, E3 Ubiquitin Protein Ligase | Reported |
| PRKG1 | Protein Kinase Cgmp-Dependent 1 | Reported |
| SGK1 | Serum/Glucocorticoid Regulated Kinase 1 | Reported |
| CLCNKB | Chloride Voltage-Gated Channel Kb | Unreported |
| CTSA | Cathepsin A | Unreported |
| CYP3A5 | Cytochrome P450 Family 3 Subfamily A Member 5 | Unreported |
| ENPEP | Glutamyl Aminopeptidase | Unreported |
| SCNN1G | Sodium Channel Epithelial 1 Gamma Subunit | Unreported |
| SLC24A3 | Solute Carrier Family 24 Member 3 | Unreported |
| THOP1 | Thimet Oligopeptidase 1 | Unreported |
| WNK1 | Wnk Lysine Deficient Protein Kinase 1 | Unreported |

functional features of the genes. Therefore, carrying out gene enrichment analysis is a vital part in exploring the high-throughput data extracted from different biological observations and experiments. This methodology helps to discover the non-predefined interaction between functional genes that significantly regulate different biological. Gene ontology analysis depicted the involvement of 125 genes in obesity and 24 genes among them were SSGs contributing to 50 percentage of total SSGs. These findings signifies the critical role of SSGs in the role of obesity. To explore more on salt related genes co-expression analysis of obesity related genes in adipose tissue was carried out. By performing co-expression analysis, we obtained 23 co-expressed SSGs with obesity related genes. Eight among the 23 co-expressed genes were not previously reported for the disease obesity via gene ontology analysis. Gene co-correlation can be explained by the fact that genes showing similar regulation/ expression patterns are frequently interconnected together than with arbitrary genes [9]. Interaction map of the unreported SSGs with obesity related genes showed stronger interactions in disease state. It is very clear from the plot (Fig 5) that majority of the co-expressed genes in the obese conditions are not co-expressed in normal conditions. The novel obesity associated SSG and their interactions supports the view that the differentially co-expressed genes are likely to get involved in numerous molecular processes resulting in adverse or balancing effects [39].

The established theory in network biology is that disease related genes existing in close physical proximity are most likely to cause diseases with similar molecular basis. In addition, in a network of disease genes, the non-disease genes are identified to have a higher tendency to interact with other disease genes [40]. Considering the theory, we looked into disease and pathway related to the prioritized gene from unreported SSGs. *WNK1* and *ENPEP* act as central hub in the network with high number co-expressed partners. In the functional enrichment

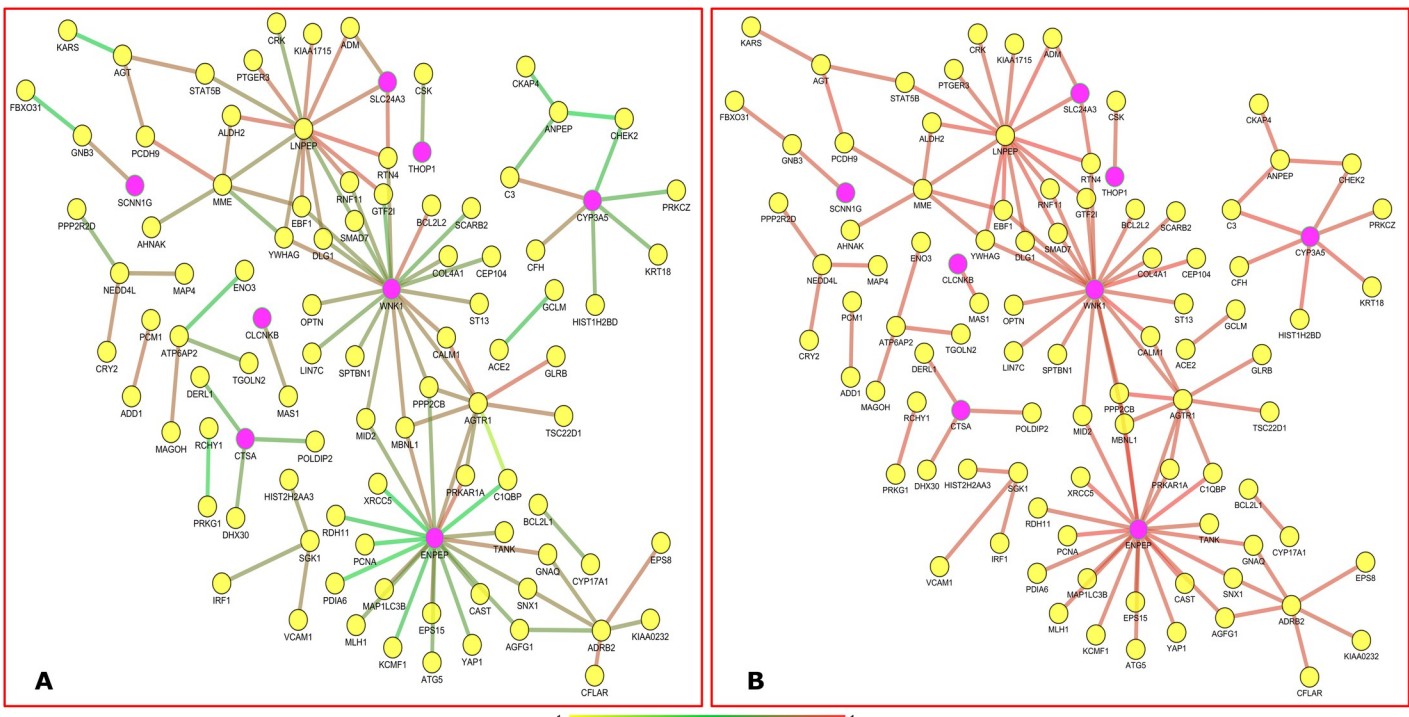

**Fig 5. The plot depicts the correlation score of gene pairs in obese and control conditions.** The color scale (-1 to +1) represents the correlation value. Higher the value higher is the correlation. (A) Represents gene-gene correlation in normal samples and (B) represents their corresponding correlation in obese condition. Pink nodes depict novel genes that are co-expressed with obesity related genes in obese condition and Yellow nodes represent obesity related genes.

data, the gene *WNK1* is reported in diseases like Diabetes Mellitus, Cardiovascular Diseases, Metabolic Syndrome X, Hyperglycemia and heart failure. These enriched diseases also show close relationship with obesity. Recent report by Ding et al., [41] in mouse model suggests *WNK1* as a novel signaling molecule involved in development of obesity. It suggests lack of Akt3 in adipocytes rises the WNK1 protein level which in turn activates SGK1 and stimulates adipogenesis through phosphorylation and inhibition of FOXO1 transcription factor, subsequently, activating the transcription of PPARg in adipocytes. Increased adipocyte results in high fat accumulation and ultimately to obesity. Thus, *WNK1*, can act as one of the potential biomarker or targets for controlling obesity. Additionally, at pathway level, *WNK1* is known to be a potent regulator of $Na^+$ and $Cl^-$ ions transport, and consequently the blood pressure. Ewout et al, (2011) describes about the role of WNKs in salt metabolism via regulating sodium, chlorine, potassium and blood pressure [42]. WNKs are involved in crucial molecular pathways via connecting hormones such as angiotensin II and aldosterone to sodium and potassium transport. *WNK1* is significantly involved in homeostasis and several biological processes regulations including and not limited to cell survival, proliferation and signaling fates. WNK1 activates sodium channel epithelial (ENaC) gene subunits SCNN1A, SCNN1B, and SCNN1D. It is also known as an activator of SGK1. In fact, by inhibiting WNK4 activity through kinase phosphorylation, WNK1 controls $Na^+$ and $Cl^-$ ions transport. Moreover, WNK1 plays a switch role-like (activation/inhibition) of the Na-K-Cl cotransporters (NKCC) respectively [43].

*ENPEP* is a member of the M1 family of endopeptidases. It is plays a role in the catabolic pathway of the renin angiotensin system which in turn is involved in regulation of blood pressure [44]. The gene is observed in Hypertensive disease which are closely associated with obesity. Currently, inhibition of *ENPEP* activity is one of procedure used to treat hypertension

**Table 6. Interactions of unreported salt sensitive genes in obese and normal condition with their corresponding co-relation score as weights.**

| Gene-1 | Gene-2 | Obese[1] | Normal[2] |
|---|---|---|---|
| WNK1 | CALM1 | 0.9439 | 0.5472 |
| ENPEP | C1QBP | 0.9417 | -0.2225 |
| ENPEP | PCNA | 0.9240 | -0.2255 |
| ENPEP | MBNL1 | 0.9156 | 0.5669 |
| WNK1 | CEP104 | 0.9145 | 0.2931 |
| ENPEP | XRCC5 | 0.9110 | -0.1939 |
| CYP3A5 | HIST1H2BD | 0.9040 | 0.2237 |
| CYP3A5 | C3 | 0.9015 | 0.6306 |
| CLCNKB | MAS1 | 0.9001 | 0.4543 |
| ENPEP | CAST | 0.8897 | 0.3039 |
| CYP3A5 | CFH | 0.8787 | 0.5056 |
| CYP3A5 | KRT18 | 0.8696 | 0.1807 |
| ENPEP | SNX1 | 0.8659 | 0.3880 |
| SLC24A3 | RTN4 | 0.8625 | 0.7491 |
| ENPEP | MID2 | 0.8569 | 0.3283 |
| ENPEP | GNAQ | 0.8480 | 0.5793 |
| WNK1 | SPTBN1 | 0.8469 | 0.3412 |
| SLC24A3 | LNPEP | 0.8434 | 0.6549 |
| WNK1 | OPTN | 0.8420 | 0.3940 |
| ENPEP | MAP1LC3B | 0.8410 | 0.4832 |
| WNK1 | ST13 | 0.8377 | 0.4385 |
| WNK1 | LIN7C | 0.8362 | 0.2726 |
| WNK1 | DLG1 | 0.8352 | 0.3487 |
| ENPEP | PPP2CB | 0.8341 | 0.2886 |
| WNK1 | BCL2L2 | 0.8338 | 0.7179 |
| ENPEP | AGFG1 | 0.8333 | 0.2477 |
| SLC24A3 | ADM | 0.8324 | 0.5586 |
| ENPEP | MLH1 | 0.8309 | 0.2835 |
| WNK1 | RNF11 | 0.8297 | 0.3692 |
| CTSA | POLDIP2 | 0.8285 | 0.1705 |
| CTSA | DERL1 | 0.8279 | 0.1225 |
| ENPEP | YAP1 | 0.8258 | 0.1928 |
| WNK1 | SMAD7 | 0.8237 | 0.2519 |
| ENPEP | KCMF1 | 0.8225 | -0.0395 |
| WNK1 | YWHAG | 0.8217 | 0.5680 |
| SCNN1G | GNB3 | 0.8216 | 0.5586 |
| ENPEP | PRKAR1A | 0.8215 | 0.7386 |
| CYP3A5 | PRKCZ | 0.8213 | 0.1144 |
| ENPEP | RDH11 | 0.8198 | -0.0770 |
| WNK1 | COL4A1 | 0.8191 | 0.3485 |
| CTSA | DHX30 | 0.8171 | 0.2648 |
| WNK1 | EBF1 | 0.8168 | 0.3779 |
| ENPEP | ATG5 | 0.8151 | 0.2386 |
| ENPEP | AGTR1 | 0.8149 | 0.4997 |
| ENPEP | EPS15 | 0.8134 | 0.5767 |
| WNK1 | AGTR1 | 0.8124 | 0.4890 |

*(Continued)*

**Table 6.** (Continued)

| Gene-1 | Gene-2 | Obese[1] | Normal[2] |
|---|---|---|---|
| WNK1 | MBNL1 | 0.8123 | 0.5277 |
| WNK1 | RTN4 | 0.8121 | 0.4261 |
| ENPEP | PDIA6 | 0.8111 | -0.2465 |
| CYP3A5 | CHEK2 | 0.8098 | 0.0108 |
| WNK1 | PPP2CB | 0.8077 | 0.4375 |
| WNK1 | SCARB2 | 0.8059 | 0.1818 |
| THOP1 | CSK | 0.8058 | 0.3290 |
| WNK1 | MID2 | 0.8047 | 0.3940 |
| ENPEP | TANK | 0.8027 | 0.3945 |
| WNK1 | GTF2I | 0.8001 | 0.1125 |

[1]obese = correlation score in obese sample

[2]normal = correlation score in normal sample

condition. Hypertension is a growing problem affecting 40% percent of adults due to the growing prevalence of obesity and diabetes in many parts of the world [45]. In addition, DNA methylation study in human adipose tissue reveals *ENPEP* as one of the differentially methylated genes associated with obesity and related traits [46]. *ENPEP* is found to be a candidate gene associated with obesity and hypertension traits in GWAS (Genome Wide Association study) studies. *ENPEP* is highly correlated with obesity related genes and also correlated with the diseases that may be comorbidity conditions of obesity. Therefore, our work provides strong evidence for *ENPEP* to be a novel gene that contributing to obesity.

*CYP3A5* plays a role in the metabolism of many drugs and other metbolites, such as steroids. *CYP3A5* is also involved in the oxidative metabolism of xenobiotics, as well as calcium channel blocking drugs and immunosuppressive drugs. *CYP3A5* is a member of the cytochrome P450 superfamily of enzymes. These proteins are monooxygenases catalyzing reactions in metabolism of drugs, cholesterol, steroids and other lipids. The main functions associated with *CYP3A5* are monooxygenase activity, iron ion binding, lipid metabolism and oxidoreductase activity [47].

Potassium-dependent sodium/calcium exchanger *(SLC24A3)* plays an important role in intracellular calcium homeostasis. It facilitates exchange of intracellular Ca$^{++}$ and K$^{+}$ ions for

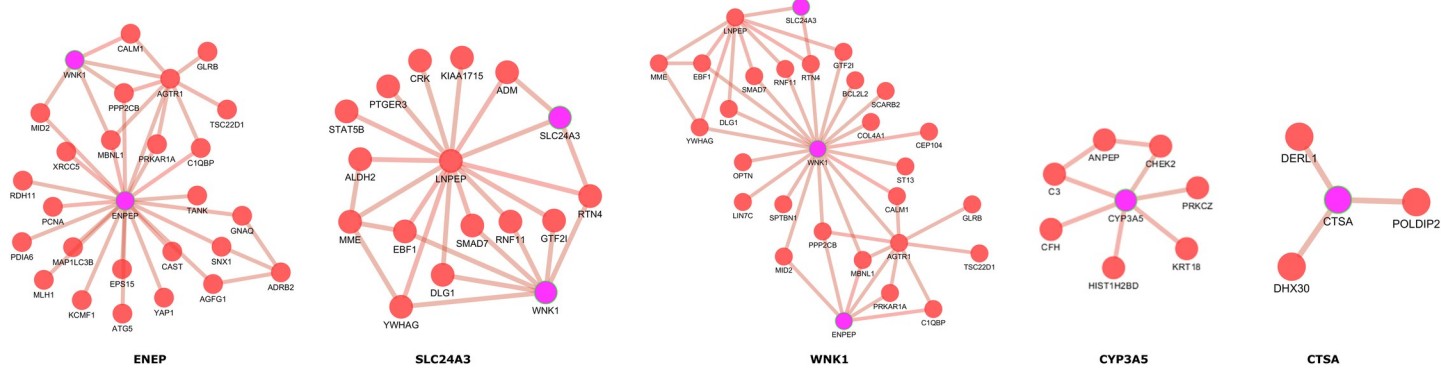

**Fig 6. The partners of prioritized unreported salt sensitive genes (ENPEP, WNK1, CYP3A5, SLC24A3 and CTSA) which are co-expressed with obesity related genes in obese condition.**

**Table 7. The traits extracted from GWAS catalogue for the unreported genes co-expressed with obesity related genes.**

| Gene | Trait | PubMed |
|---|---|---|
| ENPEP | atrial fibrillation | 17603472 |
| ENPEP | systolic blood pressure | 21572416 |
| ENPEP | diastolic blood pressure | 21572416 |
| ENPEP | cognitive impairment, cognitive decline measurement | 26252872 |
| ENPEP | eye color | 29109912 |
| ENPEP | lung carcinoma | 28604730 |
| ENPEP | metabolite measurement | 21886157 |
| ENPEP | pursuit maintenance gain measurement | 29064472 |
| SLC24A3 | chronic obstructive pulmonary disease, smoking initiation | 21685187 |
| SLC24A3 | fat body mass | 28224759 |
| SLC24A3 | matrix metalloproteinase measurement | 20031604 |
| SLC24A3 | mean platelet volume | 27863252 |
| SLC24A3 | migraine disorder | 27182965 |
| SLC24A3 | Psychosis | 24132900 |
| SLC24A3 | FEV/FEC ratio | 22424883 |
| SLC24A3 | pulse pressure measurement | 28135244 |
| SLC24A3 | unipolar depression, response to escitalopram, response to citalopram, mood disorder | 27622933 |
| SLC24A3 | Age at smoking initiation in chronic obstructive pulmonary disease | 21685187 |
| SLC24A3 | Daytime sleepiness | 28604731 |
| SLC24A3 | Matrix metalloproteinase levels | 20031604 |
| SLC24A3 | Migraine | 27182965 |
| SLC24A3 | Pulmonary function decline | 22424883 |
| SLC24A3 | Pulse pressure | 28135244 |
| SLC24A3 | QT interval | 27958378 |
| WNK1 | body mass index | 25673413 |
| WNK1 | colorectal cancer | 24836286 |
| WNK1 | eosinophil count | 27863252 |
| WNK1 | eosinophil percentage of leukocytes | 27863252 |
| WNK1 | lung carcinoma | 28604730 |
| WNK1 | smoking status measurement, lung carcinoma | 28604730 |
| WNK1 | squamous cell lung carcinoma | 28604730 |
| WNK1 | blood manganese measurement | 26025379 |
| WNK1 | Malignant epithelial tumor of ovary, response to paclitaxel | 29367611 |
| WNK1 | Stroke | 19369658 |
| CYP3A5 | Blood metabolite levels | 25898920 |
| CYP3A5 | Borderline personality disorder | 28632202 |
| CYP3A5 | Cognitive decline rate in late mild cognitive impairment | 26252872 |
| CYP3A5 | Disease progression in age-related macular degeneration | 29346644 |
| CYP3A5 | Early childhood aggressive behavior | 26087016 |
| CYP3A5 | Factor VII | 17903294 |
| CYP3A5 | Obesity-related traits | 23251661 |
| CYP3A5 | Ticagrelor levels in individuals with acute coronary syndromes treated with ticagrelor | 25935875 |
| CYP3A5 | Blood metabolite ratios | 24816252 |

extracellular sodium ions [48]. *CTSA* is a member of cathepsins family which are a group of lysosomal proteases that have a key role in cellular protein turnover. *CTSA* is not directly reported in obesity, but an analysis performed by Nadia et al., (2010) implicates cysteine

proteases cathepsins S, L, and K in complications of obesity [49]. Similarly, a study conducted by Araujo et al., (2018) reports CTSB, a member in Cathepsin family, controls autophagy in adipocytes. In obese individuals, the expression of this gene increases which in turn regulates inflammatory markers [50]. In our analysis *CTSA* is co-expressed with obesity related genes suggesting a critical role in the pathway of obesity since the members of Cathespin family plays import role in obesity. The major functions associated with *CTSA* are glycosphingolipid metabolism, protein transport and enzyme activating activity.

In GWAS analysis, the genes *CYP3A5*, *SLC24A3* and *CTSA* are observed in obesity related diseases like Hypertensive disease, Asthma, Coronary Artery Disease, Essential Hypertension, Hypertensive disease and Heart failure. We found the gene *CYP3A5* is reported as one of loci associated with obesity related traits in GWAS studies [51]. It is also associated with Factor VII and blood metabolite levels. Recent study by Takahashi et al., [52] reports the relationship of factor VII and obesity. The results propose Factor VII is an adipokine, enhanced by TNF-α or isoproterenol, which plays crucial role in the pathogenesis of obesity. *SLC24A3* and *WNK1* are mapped to traits like fat body mass and body mass index which are closely associated with obesity. This analysis of integrating GWAS studies also substantiates the possible association of novel genes identified through this study to obesity related traits and comorbidity symptoms and diseases.

We acknowledge that our strategy has some technical constraints. First, since experimentally derived protein interactions were retrieved using Bisogenet plugin. This plugin employs multiple databases of protein-protein interactions hence any interaction which has not been updated in those databases may not have been included in our study. In addition to this the insufficiency of data pertaining to certain genes in the Gene Ontology (GO) should also be considered. In order to overcome these limitations, we tried to include protein interaction based on co-expression. Overall, our research analysis has presented the effectiveness of linking genetic expression with their functional relationship in identification of obesity candidate genes. In order to demonstrate the involvement of the novel candidate genes mentioned in this study further experimental validation is required.

## Conclusions

This work systematically outlines an integrated bioinformatics pipeline for figuring out the most indispensable key signatures from the interactome Salt Sensitivity Protein Interaction Network (SS$^{PIN}$). The findings with biological relevance depict 50% of the SSGs have experimental evidences for their role in the pathogenesis of obesity. A detailed parametric downstream analysis based on biological insights, illustrated 5 candidate genes that can act as potential biomarker or target for obesity. To authenticate our results, we illustrate the possible role of ENPEP and WNK1 which appeared in the top prioritized list. Overall, our research analysis has presented the effectiveness of linking genetic expression with their functional relationship in identification of obesity candidate genes.

## Supporting information

**S1 Table. The list of samples and their characteristics used in the research analysis.**
(PDF)

**S2 Table. The list of Salt Sensitive Genes analyzed in the present study.**
(PDF)

**S3 Table. Go Annotation of Obesity Salt-Sensitivity Genes.**
(XLSX)

## Acknowledgments

This project was funded by the Deanship of Scientific Research (DSR) at King Abdulaziz University, under grant no. HiCi-63-130-35. The authors are thankful to DSR for the technical and the financial support.

## Author Contributions

**Conceptualization:** Jamal Sabir M. Sabir, Abdelfatteh El Omri, Abdulkader M. Shaikh Omar, Robert K. Jansen, Muhummadh Khan.

**Data curation:** Babajan Banaganapalli, Noor A. Shaik.

**Formal analysis:** Babajan Banaganapalli, Nada Aljuaid, Nahid H. Hajrah, Noor A. Shaik.

**Funding acquisition:** Jamal Sabir M. Sabir, Abdulkader M. Shaikh Omar.

**Investigation:** Abdelfatteh El Omri, Babajan Banaganapalli, Abdulkader M. Shaikh Omar, Noor A. Shaik, Muhummadh Khan.

**Methodology:** Abdelfatteh El Omri, Babajan Banaganapalli, Nada Aljuaid, Abdulmalik Altaf, Nahid H. Hajrah, Houda Zrelli, Mona G. Alharbi, Alawiah M. Alhebshi, Noor A. Shaik.

**Project administration:** Jamal Sabir M. Sabir, Abdelfatteh El Omri, Muhummadh Khan.

**Resources:** Abdelfatteh El Omri, Nada Aljuaid, Nahid H. Hajrah, Leila Arfaoui, Mona G. Alharbi, Alawiah M. Alhebshi.

**Software:** Babajan Banaganapalli, Leila Arfaoui, Mona G. Alharbi, Muhummadh Khan.

**Supervision:** Jamal Sabir M. Sabir, Abdelfatteh El Omri, Abdulkader M. Shaikh Omar, Houda Zrelli, Robert K. Jansen, Noor A. Shaik.

**Validation:** Nada Aljuaid, Abdulmalik Altaf, Mona G. Alharbi, Alawiah M. Alhebshi.

**Visualization:** Babajan Banaganapalli, Abdulmalik Altaf, Mona G. Alharbi, Noor A. Shaik.

**Writing – original draft:** Houda Zrelli, Ramu Elango, Muhummadh Khan.

**Writing – review & editing:** Houda Zrelli, Ramu Elango, Robert K. Jansen, Noor A. Shaik, Muhummadh Khan.

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
