## [Decision Letter · Decision Letter 0]

27 Dec 2019

PONE-D-19-29254

Unraveling the Role of Salt-Sensitive Genes in Obesity with integrated Network Biology and Co-Expression Analysis

PLOS ONE

Dear Dr Khan,

Thank you for submitting your manuscript to PLOS ONE. After careful consideration, we feel that it has merit but does not fully meet PLOS ONE’s publication criteria as it currently stands. Therefore, we invite you to submit a revised version of the manuscript that addresses the points raised during the review process.

ACADEMIC EDITOR: Please improve the quality of the images according to journal guidelines. 

We would appreciate receiving your revised manuscript by Feb 10 2020 11:59PM. To enhance the reproducibility of your results, we recommend that if applicable you deposit your laboratory protocols in protocols.io, where a protocol can be assigned its own identifier (DOI) such that it can be cited independently in the future. For instructions see: http://journals.plos.org/plosone/s/submission-guidelines#loc-laboratory-protocols

We look forward to receiving your revised manuscript.

Kind regards,

Narasimha Reddy Parine, Ph.D

Academic Editor

PLOS ONE

Journal Requirements:

2. Please amend the manuscript submission data (via Edit Submission) to include author Imran Khan.

3. Please amend your authorship list in your manuscript file to include author Leila Arfaoui.

5. Your ethics statement must appear in the Methods section of your manuscript. If your ethics statement is written in any section besides the Methods, please move it to the Methods section and delete it from any other section. Please also ensure that your ethics statement is included in your manuscript, as the ethics section of your online submission will not be published alongside your manuscript.

Reviewers' comments:

Reviewer's Responses to Questions

**Comments to the Author**

1. Is the manuscript technically sound, and do the data support the conclusions?

Reviewer #1: Yes

Reviewer #2: Yes

Reviewer #3: Yes

2. Has the statistical analysis been performed appropriately and rigorously? 

Reviewer #1: Yes

Reviewer #2: Yes

Reviewer #3: Yes

3. Have the authors made all data underlying the findings in their manuscript fully available?

Reviewer #1: Yes

Reviewer #2: Yes

Reviewer #3: Yes

4. Is the manuscript presented in an intelligible fashion and written in standard English?

Reviewer #1: Yes

Reviewer #2: Yes

Reviewer #3: Yes

5. Review Comments to the Author

Reviewer #1: The manuscript describes a study by the authors which identifies network of salt sensitivity genes and their co-expressed genes to trace the molecular pathways connected to obesity in visceral adipose tissues using gene expression and weighted protein interaction network. This work systematically outlines an integrated bioinformatics pipeline for figuring out the most indispensable key signatures from the interactome Salt Sensitivity Protein Interaction Network. Most recent bioinformatics tools and software have been employed in this investigation, hence it proves the gene's role or its association to Obesity. Touching upon a burning problem like obesity is a good analysis. The manuscript is interesting and publishable in PLOS one journal after the minor issues are addressed by authors.

1. In the introduction section, the authors have mentioned “We have also adopted another novel concept i.e. gene-gene correlation…….” replace “novel concept” with some other suitable word as the mentioned concept is not novel

2. References needed for “The statistical difference between differentially expressed genes (DEGs) was computed using unpaired t-test measure among healthy and obese samples.”

3. References needed for protein interaction databases mentioned in the section the section 2.3

4. References needed for “Next, the plugin NetworkAnalyzer, calculated the degree centrality betweenness….” Mentioned in the section 3.2

5. In the section “3.5 Co-expression analysis” Eight among the 23 co-expressed genes were not previously reported for the disease obesity via gene ontology analysis. Replace the phrase ‘gene ontology’ with suitable word like functional enrichment or gene set enrichment analysis

6. In the section, 4 Discussion,

“Local parameter DC and global parameter BC were used to dissect the complex interactome.”

The terms Local parameter and global parameter are not given any explanation in method or results section

Reviewer #2: The present study has analyzed the contribution of salt sensitivity genes (SSGs) to adiposity in obese patients. They have used a secondary gene expression data set of visceral adipose tissues for performing gene network and pathway enrichment analysis of salt sensitivity genes. They have shown that SSGs and co-expressed gene partners participate in diverse classes of metabolic pathways like those involving lipid metabolism, adipogenic pathways, renin-angiotensin system regulation, etc.

This is the first study conducted looking at the role of SSGs in adipose tissues. They used diverse systems biology methods gene correlation and topological parameters based on graph theory for expression data to identify biomarkers related adipogenesis.

I believe that their network biology provide will provide a novel association with potential biological comprehensions and support future translational assessment on SSGs and obesity. The introduction of this article provides a brief history of the problem and describes the study rationale, methods are provided in detail (including formulas), results & discussions sections are also presented well.

Overall, this article is well prepared and understandable to readers.

I recommend this study to be published in PLOS One Journal in the present form.

Reviewer #3: I would like to congratulate all the stake holders of the study, for their valuable contribution for trying to find some cues to alleviate obesity from world population. In modern world this is the major problem associated with many diseases.

6. PLOS authors have the option to publish the peer review history of their article (what does this mean?). If published, this will include your full peer review and any attached files.

Reviewer #1: No

Reviewer #2: Yes: SYED SHOEB IQBAL RAZVI

Reviewer #3: No

---

## [Author Response · Author response to Decision Letter 0]

7 Jan 2020

Reviewer #1:

The manuscript describes a study by the authors which identifies network of salt sensitivity genes and their co-expressed genes to trace the molecular pathways connected to obesity in visceral adipose tissues using gene expression and weighted protein interaction network. This work systematically outlines an integrated bioinformatics pipeline for figuring out the most indispensable key signatures from the interactome Salt Sensitivity Protein Interaction Network. Most recent bioinformatics tools and software have been employed in this investigation, hence it proves the gene's role or its association to Obesity. Touching upon a burning problem like obesity is a good analysis. The manuscript is interesting and publishable in PLOS one journal after the minor issues are addressed by authors.

1. In the introduction section, the authors have mentioned “We have also adopted another novel concept i.e. gene-gene correlation…….” replace “novel concept” with some other suitable word as the mentioned concept is not novel

Response: We have updated the manuscript with corresponding changes. The sentence is updated as “We have also used gene-gene correlation….”

2. References needed for “The statistical difference between differentially expressed genes (DEGs) was computed using unpaired t-test measure among healthy and obese samples.”

Response: We thank the reviewer for pointing out the missing reference. We have now updated the manuscript with the reference

3. References needed for protein interaction databases mentioned in the section the section 2.3

Response: We thank the reviewer for pointing out the missing reference. We have now updated the manuscript with the reference

4. References needed for “Next, the plugin NetworkAnalyzer, calculated the degree centrality betweenness….” Mentioned in the section 3.2

Response: We thank the reviewer for pointing out the missing reference. We have now updated the manuscript with the reference

5. In the section “3.5 Co-expression analysis” Eight among the 23 co-expressed genes were not previously reported for the disease obesity via gene ontology analysis. Replace the phrase ‘gene ontology’ with suitable word like functional enrichment or gene set enrichment analysis

Response: We have now updated the article by replacing gene ontology with functional enrichment. 

6. In the section, 4 Discussion, “Local parameter DC and global parameter BC were used to dissect the complex interactome.” The terms Local parameter and global parameter are not given any explanation in method or results section

Response: We thank the reviewer for pointing out the missing information. We have now updated the manuscript with the information about local and global parameters

Reviewer #2:

The present study has analyzed the contribution of salt sensitivity genes (SSGs) to adiposity in obese patients. They have used a secondary gene expression data set of visceral adipose tissues for performing gene network and pathway enrichment analysis of salt sensitivity genes. They have shown that SSGs and co-expressed gene partners participate in diverse classes of metabolic pathways like those involving lipid metabolism, adipogenic pathways, renin-angiotensin system regulation, etc.

This is the first study conducted looking at the role of SSGs in adipose tissues. They used diverse systems biology methods gene correlation and topological parameters based on graph theory for expression data to identify biomarkers related adipogenesis. 

I believe that their network biology provide will provide a novel association with potential biological comprehensions and support future translational assessment on SSGs and obesity. The introduction of this article provides a brief history of the problem and describes the study rationale, methods are provided in detail (including formulas), results & discussions sections are also presented well. 

Overall, this article is well prepared and understandable to readers. I recommend this study to be published in PLOS One Journal in the present form.

Response: We thank reviewer for recommending our publication.

Reviewer #3:

I would like to congratulate all the stake holders of the study, for their valuable contribution for trying to find some cues to alleviate obesity from world population. In modern world this is the major problem associated with many diseases.

Response: We thank reviewer for recommending our publication.

---

## [Editor Report · Decision Letter 1]

15 Jan 2020

Unraveling the Role of Salt-Sensitivity Genes in Obesity with integrated Network Biology and Co-Expression Analysis

PONE-D-19-29254R1

Dear Dr. Khan,

We are pleased to inform you that your manuscript has been judged scientifically suitable for publication and will be formally accepted for publication once it complies with all outstanding technical requirements.

With kind regards,

Narasimha Reddy Parine, Ph.D

Academic Editor

PLOS ONE

---

## [Editor Report · Acceptance letter]

22 Jan 2020

PONE-D-19-29254R1 

Unraveling the Role of Salt-Sensitivity Genes in Obesity with integrated Network Biology and Co-Expression Analysis 

Dear Dr. Khan:

I am pleased to inform you that your manuscript has been deemed suitable for publication in PLOS ONE. Congratulations! Your manuscript is now with our production department. 

With kind regards,

on behalf of

Dr. Narasimha Reddy Parine 

Academic Editor

PLOS ONE